# Behavioral predictors associated with HIV screening needs in gay Korean men during the COVID-19 pandemic

**Rang Hee Kwon**[1], **So-Hyun Kim**[1], **Minsoo Jung**[1,2]*

**1** Department of Health Science, Dongduk Women's University, Seoul, South Korea, **2** Center for Community-Based Research, Dana-Farber Cancer Institute, Boston, Massachusetts, United States of America

* mins.jung@gmail.com, mj748@dongduk.ac.kr

## Abstract

During the pandemic, the capacity of medical resources focused on testing, diagnosing and treating COVID-19 has severely limited public access to health care. In particular, HIV screening, for which homosexual males in Korea received free and anonymous testing at public health centers, was completely halted. This study investigated behavioral predictors related to the HIV screening needs of Korean male homosexuals during the pandemic. Data were collected by conducting a web survey targeting members of the largest homosexual portal site in Korea with support from the National Research Foundation of Korea (n = 1,005). The key independent variables are COVID-19-related characteristics and sexual risk behavior. The moderating variable is health information search behavior, and the dependent variable is the need for HIV screening. For a statistical analysis, a hierarchical multiple logistic regression analysis was conducted while controlling for potential confounding variables. According to the results of this study, the need for HIV screening was 0.928 times lower for older people (p<0.05, 95% CI = 0.966–0.998). However, if the respondent had a primary partner, the need for HIV screening was 1.459 times higher (p<0.01, 95% CI = 1.071–1.989). In addition, the need for screening was 1.773 times higher for those who preferred anal intercourse (p<0.01, 95% CI = 1.261–2.494) and 2.034 times higher (p<0.01, 95% CI = 1.337–3.095) if there was a history of being diagnosed with an STD. Finally, health information-seeking behavior was marginally significant. This study revealed that male Korean homosexuals who were young, preferred anal sex with a primary partner, and who had a history of a sexually transmitted disease had a high need for HIV screening at public health centers. They are more likely to be susceptible to HIV infection because they are usually consistent with gay men, characterized by risky behavior. Therefore, an intervention strategy that provides health information using a communication campaign is needed.

**Data Availability Statement:** Data cannot be shared publicly because of the policy of the National Research Foundation of Korea Grant funded by the Korean Government (NRF-

2022R1F1A1062998). Data are available from the Institutional Review Board of Dongduk Women's University (DDWU2206-02) for researchers who meet the criteria for access to confidential data. The person in charge of Dongduk Women's University's IRB is Researcher Seol Chae-yeon. The phone number is +82-2-940-4236 and the email address is tjfcodus@dongduk.ac.kr. Upon your request, she will provide you with data from this study after approval by NRF.

**Funding:** This work was supported by the National Research Foundation of Korea Grant funded by the Korean Government (NRF-2022R1F1A1062998; PI: Prof. Dr. Minsoo Jung). The funders had no role in study design, data collection and analysis, decision to publish, or preparation of the manuscript.

**Competing interests:** The authors declared no potential conflicts of interest with respect to the research, authorship, and/or publication of this article.

## Introduction

In 2021, 38,400,000 people worldwide were living with human immunodeficiency virus (HIV), with about 1,500,000 newly infected and about 650,000 dying from AIDS [1]. There are many ways to curb the spread of HIV infection, and MSM (men who have sex with men) being tested for HIV frequently is one way to reduce the infection rate [2]. However, COVID-19 has reduced healthcare utilization overall [3–10]. However, it is usually assumed that this is not a decrease in the incidence of the disease itself. According to previous studies, it was found that necessary medical use was suppressed due to a lack of medical resources, reduced accessibility due to medical examinations by interview, and an increased risk of COVID-19 infection as perceived by the patient. However, the number of patients with chronic diseases increased [3, 11, 12].

In Korea, one or more public health centers are established in each city, county, and district based on Article 10 of the Regional Health Act. Currently, approximately 250 public health centers nationwide provide health education and health services to promote health and prevent disease for local residents, including anonymous HIV screening tests. However, after the COVID-19 pandemic was declared, free anonymous HIV testing provided by public health centers was discontinued, and access to HIV screening by MSM was significantly limited. In fact, according to a survey titled *The Impact of COVID-19 on Access to and Supply of HIV Drugs in the Asia-Pacific Region* published by Gilead Sciences in December of 2020, 48.34% of the HIV infection risk group in Korea reported fewer HIV tests due to COVID-19, and 59.6% responded that the frequency of their hospital visits decreased (41.72%) or that they had not yet visited the hospital (17.88%) [13]. During the pandemic, all public health centers in Korea mobilized human resources for COVID-19 screening. As a result, there were only three iSHAP (Ivan Stop HIV/ADIS; iSHAP) locations nationwide that could provide free, anonymous rapid HIV testing. However, despite the fact that public health centers, which have played a crucial role in HIV screening in Korea, have been suspended for more than two years, there has been no study of the unmet needs of MSM or new HIV incidence rates. As an exploratory study for this purpose, we explored behavioral predictors related to HIV screening needs by MSM during the COVID-19 pandemic.

The epidemiological characteristics of those newly infected with HIV in Korea show that they are mostly males in their 20s and those with a homosexual orientation who prefer high-risk sexual behaviors [14]. In 2020, when COVID-19 had just begun, it is known that sexual risk behaviors among MSM were on the decline due to strong quarantine policies [15], but in May of 2020, what has been termed an Itaewon group infection occurred in Korea. At a time when the number of confirmed cases of COVID-19 was very low, as a group infection occurred at a favorite club of MSM, comments reporting that MSM did not maintain a physical distance well and would not stop engaging in sexually risk behavior spread on social media [16]. This may be a stereotype due to social stigma or discrimination [17]. However, continuing to engage in sexually risky behavior with others in a situation where state-provided HIV screening services are suspended not only increases the likelihood of exposure to HIV/STI [18] but also increases the likelihood of a COVID-19 infection [19, 20]. Therefore, we need to know how policies such as physical distancing have affected sexual behavior in MSM during the COVID-19 pandemic.

Previous studies have shown that people living with HIV are more concerned about prevention and health care during the COVID-19 pandemic than people who are not infected [21]. In addition, those who had undergone screening for HIV/STI prior to COVID-19 or who perceived themselves to be at a high risk of infection used these screening services after the outbreak of COVID-19 [18]. On the other hand, it is known that the lower the level of knowledge about HIV, the weaker the intention to screen for HIV [22]. Meanwhile, these predictive

factors related to HIV screening can be mediated by health information. Individuals tend to seek health information to improve their health status when their health condition deteriorates, and the perceived health status affects behavioral intention [23]. For example, MSM, who may be exposed to HIV/STI at any time, are likely to seek health information actively to protect themselves. HISB (health information-seeking behavior) is a health communication practice to meet the need for health information [24]. Although these various variables may be associated with the HIV screening needs of MSM, there has been little modeling or related research on them. In order to understand the intervention effect of HIV screening services, it is necessary to calculate the HIV infection rate, but the denominator cannot be accurately determined due to the nature of this hidden population. However, even in countries with low HIV prevalence, such as Korea, sexual behavior monitoring is necessary [25], especially during COVID-19, when HIV screening has been suspended and physical distancing has been enforced. Therefore, this study explored behavioral predictors related to the HIV screening needs of Korean MSM during the COVID-19 pandemic. This study can serve as a policy basis related to the provision of appropriate HIV screening services to MSM in the event of a new infectious disease pandemic.

## Methods

### Sample

The research subjects of this study are valid and active members of Ivan City (https://ivancity.com/bbs/login.php). Ivan City, which is Korea's largest homosexual community portal site, is operated by LGBT KOREA and has been operating since 1999. There are approximately 360,000 members of Ivan City in total, and according to confidential data from the Korea Federation for HIV/AIDS Prevention, 95% of the members are MSM. However, there are multiple accounts, dormant accounts, and blocked accounts among the total membership. This study determined the number of study subjects through the following two criteria.

First, it is a theoretical estimation considering the characteristics of the hidden population. Denominator information is required to calculate the appropriate number of study subjects, but little is known about the size of the denominator in that the study subjects in this study are a hidden population. However, it is estimated that the sexual minority population, including homosexuals, bisexuals, and transgender people, typically accounts for about 2.2–5.6% of the total population of a society [26]. However, it is difficult to apply this figure directly to Korea. To calculate prevalence, it is necessary to define a population and obtain a representative sample through random sampling. If the number of study subjects is sufficiently large, the approximate required number of samples can be calculated by substituting the sampling error and expected prevalence using the formula below.

$$n \geq \left( \frac{2Z_{1-a/2}\sqrt{p(1-p)}}{2*Sampling\ Error} \right)^2$$

Given that HIV is transmitted through sexual contact, we assumed a maximum prevalence rate of 30% by referring to the domestic prevalence of sexually transmitted diseases associated with MSM (i.e., syphilis, gonorrhea, chlamydia) and estimated the value so that the sample number n reached a prevalence rate of 0.5 according to the above formula. Assuming a confidence interval of 95%, the number of samples required for the G*Power program is 1,063. Accordingly, the population size for this study was set to 1,000.

Second, it is a practical estimate considering the web survey method. Through a business agreement with the Ivan City management team, we randomly extracted 10,000 IDs, which

account for about 5% of active accounts. They were provided with a link to the web survey through Hankook Research Inc., a consigned research institute, and we received 1,005 responses, or about 10% (n = 1,005). Hankook Research, the research institute with the largest sales volume in Korea, was in charge of the online research part of this study through a consignment contract with the principal investigator. All information collected by Hankook Research is de-identified for the protection of the respondents. On the other hand, in the web survey, we utilized a double-confirmation process to exclude respondents who were 1) female, 2) under the age of 19 or over the age of 60, and 3) who had no experience of sexual contact with the same sex through a subject suitability question.

We collected data from July 1, 2022 to July 30, 2022, as HIV anonymous free screening was resumed at public health centers in Korea from July 1, 2022. Respondents accessed the Ivan City website, logged in, read the note about this survey, and then clicked the external link provided by Hankook Research to participate in the online survey if they were interested in this survey. The time taken to complete the self-administered questionnaire was approximately 20 minutes. The reward for the respondent was a coupon worth 5,000 Korean won (about 3.8 USD) to Ivan City's account without any collection of identification information such as mobile phone numbers. Therefore, anonymity was guaranteed while securing the reliability and validity of the responses.

## Analytical framework

The analysis framework of this study is based on the MSM behavioral indicators of UNAIDS and FHI (Family Health International), such as the type of sexual partnership, anal sex, and condom use (Fig 1). For example, the risk of HIV infection is higher if one enjoys anal sex with a one-time partner or does not use a condom; therefore, these MSM may have a high need for HIV screening [27]. The survey instruments and self-administered questionnaire of this study were verified for validity and reliability in "Developing the behavioral monitoring survey system for the high HIV risk group", a project supported by the Korea Federation for HIV/Prevention in 2011.

## Measures

**Dependent variables.** The dependent variable of this study was whether or not there was a need for HIV screening during the COVID-19 pandemic. We asked respondents if they had

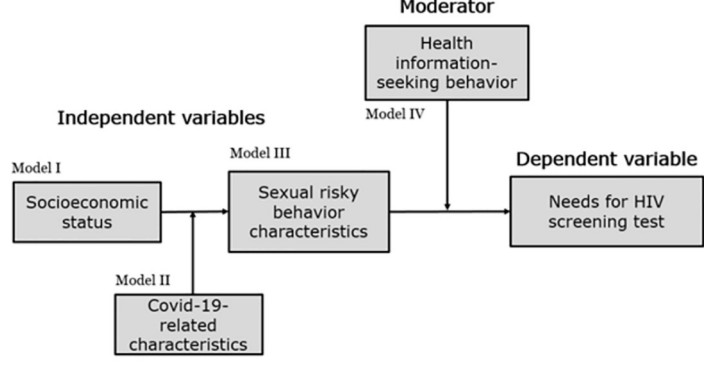

**Fig 1. Analytical frame of this study.**

ever wanted to take an HIV test from a public health center since the COVID-19 outbreak in 2020 but did not get one, and the response was "yes" or "no."

**Independent variables.**   The independent variables in this study are COVID-19-related variables and variables related to sexual behavior.

First, as variables related to sexual behavior, condom use, the type of sexual partnership (primary/casual), anal sex, sexually transmitted infection, and HIV infection status were measured. When asked about condom use, "How often do you usually use a condom during sexual intercourse?" The responses were "every time (used 100%)" (1), "frequently used (used between 50% and 90%)" (2), "occasionally used (used between 10% and 50%)" (3), and "mostly not used (less than 10% used)" (4). The question asking about the types of sexual partners was "Do you have a sex partner of the same sex with whom you are in a primary relationship?" Responses were divided into "yes" (1) and "no" (0). The question on anal sex was "Do you prefer anal sex?" Responses were divided into "prefer" (1) and "don't like" (0). The question on sexually transmitted infections was "Have you been diagnosed with or treated for a sexually transmitted disease in the past year?" Responses were divided into "yes" (1) and "no" (0). Finally, the question on HIV infection was "Are you infected with HIV?" Responses were divided into "yes" (1) and "no" (0).

Second, COVID-19-related variables measured completion of vaccination and corona infection. The question on completion of vaccination was "How many times have you been vaccinated with a Corona-19 vaccine?" Responses were divided into "never vaccinated" (1), "one" (2), "two" (3), and "three or more" (4). We classified the type of vaccine received by the respondent and reclassified the data into "inoculation complete" (1) and "inoculation incomplete" (0), after which we input them into the model. The question pertaining to COVID-19 infection was "Have you ever been infected with COVID-19 (even if you are currently infected)?" Responses were divided into "yes" (1) and "no" (0).

**Moderators.**   Health-information-seeking behavior (HISB) was set as a moderating variable to determine whether health behavior affects the strength of a significant relationship between independent and dependent variables. The question on HISB was "How actively do you usually seek health information?" Responses were categorized into "I search very hard" (5), "I tend to search hard" (4), "Normal amount" (3), "I tend not to search" (2), and "I never search" (1).

**Potential confounders.**   Each respondent's age, educational attainment, and income as potential confounding variables were controlled in the model.

## Statistical analyses

First, a descriptive statistical analysis of the general characteristics of the respondents was conducted. Second, the chi-square test was used to examine the relationship between HIV screening needs and the major variables of this study. Third, by performing a hierarchical multiple logistic regression analysis, predictors affecting HIV screening needs were identified. All statistical analyses were performed using SPSS version 25.0 (IBM Corp., Armonk, NY, USA).

## Ethics statement

Sampling and recruitment procedures complied with the research ethics guidelines of the National Research Foundation of Korea and were approved by the institutional review board of Dongduk Women's University (DDWU2206-02). In order to protect vulnerable research participants, we obtained informed consent from the participants before conducting the survey. Hankook Research Inc., which was responsible for data collection for this study, received responses to the survey after obtaining written informed consent from each of 1,005

respondents. And the entire process of data collection was reviewed and approved by Dong-duk Women's University IRB.

## Results

### Descriptive statistics of the sample

The general characteristics of the respondents are as follows (Table 1). In terms of age, those 19 to 20 years of age accounted for most of the sample, at 39.9%, and in terms of educational attainment, college graduates accounted for the most at 55.1%. In terms of income, amounts of 14,001 to 28,000 USD were most commonly reported at 21.3%. Regarding COVID-19 vaccination, 6.5% reported incomplete and 93.5% reported complete vaccination. For the question on COVID-19 infection, 59.7% were non-infected and 40.3% had been infected. Among the characteristics of sexual behavior, 34.2% of respondents reported that they often use condoms. Approximately 59.5% of the respondents reported that they did not have a primary sex partner, and 40.5% of those who responded to the questionnaire reported that they did. Regarding anal sex, 41.9% disliked it and 58.1% preferred it. Respondents who had been diagnosed with an STD within one year accounted for 14.0% of the sample. Approximately 6.6% of the respondents reported that they were HIV-positive and 24.5% stated that they needed HIV screening. In other words, they had this unmet need during the COVID-19 pandemic. About 51.9% of the respondents reported that their health information search behavior was average.

### Associations between COVID-19-related behaviors, sexually risky behaviors, and an unmet need for HIV testing

The results of examining the associations between COVID-19-related characteristics and sexual behavioral characteristics and the need for HIV screening are as follows (Table 2). MSM had a higher need for HIV screening when they had a primary sex partner (= 12.206, p<0.001), when they preferred anal intercourse (= 19.968, p<0.001), and when they were not diagnosed with a STD (= 15.251, p<0.001).

### Behavioral predictors of an unmet need for HIV testing among MSMs

The results of hierarchical multiple logistic regression analyses on predictive factors affecting HIV screening needs are as follows (Table 3). According to Model I, the need for HIV screening in MSM was 0.928 times lower with age (p<0.05, 95% CI = 0.966–0.998). COVID-19-related characteristics in Model II were not statistically significantly associated with HIV screening needs. In Model III, in which sexual behavior characteristics were added in model II, the need for HIV screening by MSM was 1.459 times higher when there was a primary sex partner (p<0.01, 95% CI = 1.071–1.989) and the need was 1.773 times higher when they preferred for anal sex (p<0.01, 95% CI = 1.261–2.494). In addition, the need by MSM for HIV screening was 2.034 times higher if they had a history of STD diagnosis (p<0.01, 95% CI = 1.337–3.095). However, the need was 0.477 times lower (p<0.05, 95% CI = 0.242–939) if infected with HIV. Finally, according to Model IV, which additionally inputs the HISB variable of the respondent, HISB was found to be marginally significant, and when more health information was sought, there was a higher need for HIV screening by 1.190 times (p = 0.054, 95% CI = 0.997–1.420).

## Discussion

During the COVID-19 pandemic, the concentration of medical resources on screening and treatment for the COVID-19 virus has limited access to HIV-related public health services. We

**Table 1. General characteristics of the sample (N = 1005).**

| Categories | | | Frequency (n) | Percentage (%) |
|---|---|---|---|---|
| Socioeconomic status | Age (y) | 19~29 | 401 | 39.9 |
| | | 30~39 | 302 | 30.0 |
| | | 40~49 | 202 | 20.1 |
| | | 50~59 | 100 | 10.0 |
| | Educational attainment | Middle school or less | 10 | 1.0 |
| | | High school | 299 | 29.8 |
| | | College | 554 | 55.1 |
| | | Post-graduate | 142 | 14.1 |
| | Annual income (USD) | < 7000 | 176 | 17.5 |
| | | 7001~14000 | 52 | 5.2 |
| | | 14001~21000 | 205 | 20.4 |
| | | 21001~28000 | 214 | 21.3 |
| | | 28001~35000 | 128 | 12.7 |
| | | 35001~42000 | 77 | 7.7 |
| | | 42001~48000 | 36 | 3.6 |
| | | 48001~56000 | 43 | 4.3 |
| | | 56001~63000 | 21 | 2.1 |
| | | 63001~70000 | 20 | 2.0 |
| | | ≥70001 | 33 | 3.3 |
| COVID-19-related characteristics | Vaccination status | Not vaccinated | 65 | 6.5 |
| | | Fully vaccinated (2nd dose) | 940 | 93.5 |
| | Infection status | Never | 600 | 59.7 |
| | | Ever | 405 | 40.3 |
| Sexual behavior characteristics | Condom use | Every time (used 100%) | 288 | 28.7 |
| | | Frequently used (50%~90%) | 328 | 32.6 |
| | | Occasionally used (10~50%) | 196 | 19.5 |
| | | Almost not used (less than 10%) | 148 | 14.7 |
| | | Non response | 45 | 4.5 |
| | Primary sex partner | Have not | 598 | 59.5 |
| | | Have | 407 | 40.5 |
| | Anal sex | Not preferred | 421 | 41.9 |
| | | Preferred | 584 | 58.1 |
| | STD infection | Never | 864 | 86.0 |
| | | Ever | 141 | 14.0 |
| | HIV status | Negative | 939 | 93.4 |
| | | Positive | 66 | 6.6 |
| Needs for HIV screening during the pandemic | | No | 759 | 75.5 |
| | | Yes | 246 | 24.5 |
| Health information-seeking behavior | | Never search | 36 | 3.6 |
| | | Not to search | 96 | 9.6 |
| | | Normal | 522 | 51.9 |
| | | Search hard | 255 | 25.4 |
| | | Search very hard | 96 | 9.6 |
| Total | | | 1005 | 100.0 |

Equivalized household annual income; USD $1 = KRW(Korean won) 1,412.00 (October, 2022)

**Table 2. Associations between COVID-19-related behaviors, sexual risky behaviors, and HIV test unmet needs (Frequency, %).**

| Categories | | | HIV screening needs | | Total |
|---|---|---|---|---|---|
| | | | No | Yes | |
| Socioeconomic status | Age (y) | 19~29 | 289 (72.1) | 112 (27.9) | 401 (100.0) |
| | | 30~39 | 230 (76.2) | 72 (23.8) | 302 (100.0) |
| | | 40~49 | 158 (78.2) | 44 (21.8) | 202 (100.0) |
| | | 50~59 | 82 (82.0) | 18 (18.0) | 100 (100.0) |
| | | $\chi^2 = 0.126$ | | | |
| | Educational attainment | Middle school or less | 5 (50.0) | 5 (50.0) | 10 (100.0) |
| | | High school | 222 (74.2) | 77 (25.8) | 299 (100.0) |
| | | College | 422 (76.2) | 132 (23.8) | 554 (100.0) |
| | | Post-graduate | 110 (77.5) | 32 (22.5) | 142 (100.0) |
| | | $\chi^2 = 4.203$ | | | |
| | Annual income (USD) | < 7000 | 129 (73.3) | 47 (26.7) | 176 (100.0) |
| | | 7001~14000 | 40 (76.9) | 12 (23.1) | 52 (100.0) |
| | | 14001~21000 | 148 (72.2) | 57 (27.8) | 205 (100.0) |
| | | 21001~28000 | 160 (74.8) | 54 (25.2) | 214 (100.0) |
| | | 28001~35000 | 104 (81.3) | 24 (18.8) | 128 (100.0) |
| | | 35001~42000 | 58 (75.3) | 19 (24.7) | 77 (100.0) |
| | | 42001~48000 | 30 (83.3) | 6 (16.7) | 36 (100.0) |
| | | 48001~56000 | 30 (69.8) | 13 (30.2) | 43 (100.0) |
| | | 56001~63000 | 18 (85.7) | 3 (14.3) | 21 (100.0) |
| | | 63001~70000 | 14 (70.0) | 6 (30.0) | 20 (100.0) |
| | | ≥70001 | 28 (84.8) | 5 (15.2) | 33 (100.0) |
| | | $\chi^2 = 9.115$ | | | |
| COVID-19-related characteristics | Vaccination status | Not vaccinated | 48 (73.8) | 17 (26.2) | 65 (100.0) |
| | | Fully vaccinated (2nd dose) | 711 (75.6) | 229 (24.4) | 940 (100.0) |
| | | $\chi^2 = 0.106$ | | | |
| | Infection status | Never | 466 (77.7) | 134 (22.3) | 600 (100.0) |
| | | Ever | 293 (72.3) | 112 (27.7) | 405 (100.0) |
| | | $\chi^2 = 3.703$ | | | |
| Sexual behavior characteristics | Condom use | Every time (used 100%) | 228 (79.2) | 60 (20.8) | 288 (100.0) |
| | | Frequently used (50%~90%) | 231 (70.4) | 97 (29.6) | 328 (100.0) |
| | | Occasionally used (10~50%) | 146 (74.5) | 50 (25.5) | 196 (100.0) |
| | | Almost not used (less than 10%) | 116 (78.4) | 32 (21.6) | 148 (100.0) |
| | | $\chi^2 = 7.268$ | | | |
| | Primary sex partner | Have not | 475 (79.4) | 123 (20.6) | 598 (100.0) |
| | | Have | 284 (69.8) | 123 (30.2) | 407 (100.0) |
| | | $\chi^2 = 12.206$ *** | | | |
| | Ana sex | Not preferred | 348 (82.7) | 73 (17.3) | 421 (100.0) |
| | | Preferred | 411 (70.4) | 173 (29.6) | 584 (100.0) |
| | | $\chi^2 = 19.968$ *** | | | |
| | STD infection | Never | 671 (77.7) | 193 (22.3) | 864 (100.0) |
| | | Ever | 88 (62.4) | 53 (37.6) | 141 (100.0) |
| | | $\chi^2 = 15.251$ *** | | | |
| | HIV status | Negative | 707 (75.3) | 232 (24.7) | 939 (100.0) |
| | | Positive | 52 (78.8) | 14 (21.2) | 66 (100.0) |
| | | $\chi^2 = 0.407$ | | | |

(*Continued*)

**Table 2.** (Continued)

| Categories | | HIV screening needs | | Total |
|---|---|---|---|---|
| | | **No** | **Yes** | |
| Health information-seeking behavior | Never search | 31 (86.1) | 5 (13.9) | 36 (100.0) |
| | Not to search | 76 (79.2) | 20 (20.8) | 96 (100.0) |
| | Normal | 395 (75.7) | 127 (24.3) | 522 (100.0) |
| | Search hard | 190 (74.5) | 65 (25.5) | 255 (100.0) |
| | Search very hard | 67 (69.8) | 29 (30.2) | 96 (100.0) |
| $\chi^2 = 4.726$ | | | | |

***p < .001

explored behavioral predictors associated with the HIV screening needs of Korean MSM in order to understand how much HIV screening was required by MSM with certain characteristics in the context of the COVID-19 pandemic. The major findings are presented below.

First, this study revealed that during the COVID-19 period, MSM who preferred anal sex had a higher need for HIV screening than MSM who did not prefer it. We also found that the

**Table 3. Results of hierarchical multiple logistic regression analyses for behavioral predictors of HIV screening test unmet needs among MSMs.**

| | | Model I | | Model II | | Model III | | Model IV | |
|---|---|---|---|---|---|---|---|---|---|
| | | **OR** | **95% CI** | **OR** | **95% CI** | **OR** | **95% CI** | **OR** | **95% CI** |
| Socioeconomic status | Age | 0.982* | 0.966–0.998 | 0.983* | 0.967–0.999 | 0.982* | 0.965–0.999 | 0.980* | 0.964–0.997 |
| | Education | 0.897 | 0.717–1.123 | 0.892 | 0.712–1.118 | 0.872 | 0.690–1.103 | 0.863 | 0.682–1.092 |
| | Income | 0.985 | 0.925–1.050 | 0.985 | 0.924–1.050 | 0.995 | 0.931–1.063 | 0.989 | 0.925–1.057 |
| COVID-19-related characteristics | Fully vaccinated (Ref.: Not vaccinated) | | | 0.974 | 0.546–1.738 | 0.873 | 0.473–1.610 | 0.848 | 0.459–1.566 |
| | COVID-19 ever infected (Ref.: Never infected) | | | 1.289 | 0.961–1.728 | 1.229 | 0.903–1.671 | 1.231 | 0.905–1.676 |
| Sexual behavior characteristics | Condom use | | | | | 0.898 | 0.771–1.045 | 0.909 | 0.780–1.060 |
| | Primary sex partner (Ref.: None) | | | | | 1.459* | 1.071–1.989 | 1.441* | 1.057–1.967 |
| | Anal sex (Ref.: Not preferred) | | | | | 1.773** | 1.261–2.494 | 1.775** | 1.260–2.499 |
| | STD ever infected (Ref.: Never infected) | | | | | 2.034** | 1.337–3.095 | 2.038** | 1.338–3.106 |
| | HIV-infected (Ref.: Not infected) | | | | | 0.477* | 0.242–0.939 | 0.462* | 0.234–0.915 |
| Health information-seeking behavior | | | | | | | | 1.190† | 0.997–1.420 |
| Nagelkereke $R^2$ | | 0.062 | | 0.116 | | 0.273 | | 0.288 | |

Values are presented as odds ratio (95% confidence interval).

Ref, reference.

*p < .05,

** p < .01,

†p < .1.

need for HIV screening was higher in the group with the experience of being diagnosed with a STD and that the need for HIV screening was higher among people living with HIV than among non-infected MSMs. These results are similar to previous studies that found that the more people pursue risky sexual behaviors, the higher the availability of HIV testing [17, 19, 28]. In other words, the previously known risk behaviors of MSMs were factors that increased the need for HIV screening, even during the COVID-19 period. Therefore, if there is no link between quarantine measures such as physical distancing due to the COVID-19 pandemic and sexual risk behaviors by MSM, public HIV screening services should continue to be provided despite COVID-19 [29–31]. Even if COVID-19 has reduced health care accessibility overall, completely suspending public services for HIV screening could lead to instances of the hidden transmission of HIV or could limit opportunities for early treatment. Approximately 24.5% of the respondents in this study reported that they had unmet needs for HIV screening during COVID-19. Therefore, it is desirable for the Korean government to continue to provide anonymous HIV screening services even in a post-pandemic situation at one quarter of public health centers by region, in proportion to the population.

Second, this study showed that COVID-19 vaccination and COVID-19 infection in MSM did not affect their need for HIV screening. Vaccination is necessary to protect oneself directly and to reduce the spread of the disease [32]. According to previous studies, the intention to receive the COVID-19 vaccine was high in the perceived susceptibility group [33]. In addition, the majority of MSM recognized that there was a high risk of the transmission of COVID-19 due to sexual contact, and the more compliant the COVID-19 quarantine measures were, the greater the likelihood of vaccination [28]. Like HIV, motivation to gain protection from COVID-19 and self-efficacy may have played a role here. Because COVID-19 and HIV have a similar infection mechanism in that they are transmitted based on human contact, the two aforementioned behaviors may be related in terms of disease prevention, but in this study, these two behaviors were independent of each other. However, at the time of our survey, 93.5% of respondents had completed their COVID-19 vaccination. Therefore, a more elaborate follow-up study is needed in this regard because, as in the Itaewon mass infection mentioned in the introduction, MSM are likely to be vulnerable to both HIV and new infectious diseases stemming from a pandemic. Therefore, from an intersectionality perspective, it is necessary to examine whether COVID-19 was more disproportionately threatening to sexual minorities.

Third, this study found that MSM with a primary sex partner had a higher need for HIV screening than MSM without a primary sex partner. These results may be contrary to previous studies that found that MSM with casual sex partners have high HIV screening utilization [28]. In the COVID-19 situation, the tendency to meet new partners was temporarily suppressed due to quarantine measures such as physical distancing [34, 35]. However, due to limited access to HIV screening services during COVID-19, the mere act of having sex with a primary partner may have triggered the intent to test. Because the results of this study may be unique to Korea, additional research is needed.

Fourth, this study found that HISB can play a mediating role in increasing the need for HIV screening by MSM. In order for individuals to prevent or effectively treat HIV, they must be able actively to seek out and apply health information. According to previous studies, the greater the HISB orientation is, the greater the treatment adherence and self-efficacy will be, which can be helpful in managing HIV [36]. MSM lacking HIV-related knowledge and awareness tend to have lower rates of HIV screening [37]. Therefore, in order to mitigate this problem, it is requested in the era of COVID-19 to develop an intervention strategy that provides health information by utilizing the HISB characteristics of MSM.

## Limitations

This study had a number of potential limitations. First, this result cannot be generalized to all Korean MSMs due to internet accessibility and selection bias in the respondents. In Korean society, sexual minorities have long suffered from social stigma and discrimination; thus, they do not reveal themselves easily and have difficulty securing representation. Second, because there is a gap between intention and behavior [38], the unmet need for HIV screening may not lead to an actual screening test. Although most smokers know that smoking is harmful to them, they smoke; they also know that vaccination prevents infectious diseases, but not all get vaccinated. Therefore, regarding the factors that actually determine HIV screening behavior by MSM, follow-up studies that apply rigorous epidemiological designs, such as natural experiments, are needed.

## Conclusion

This study investigated behavioral predictors related to HIV screening needs in Korean MSM during the COVID-19 pandemic. According to the results of the study, MSM who were young, preferred anal sex with their primary partner, and experienced sexually transmitted diseases had a high need for HIV screening provided by public health centers. These characteristics are nearly identical to those of MSM, who mainly used public health centers before COVID-19. Therefore, despite quarantine policies such as physical distancing, it is assumed that risky behaviors by MSM were generally similar before and after COVID-19. Therefore, a nationwide suspension of HIV screening in public health centers during a pandemic would be unreasonable as it would increase the vulnerability of MSM to HIV infection. An intervention strategy is needed partially to provide HIV screening services by region and to provide related information to MSM through a health communication campaign.

## Supporting information

**S1 File.**
(PDF)

## Author Contributions

**Conceptualization:** Rang Hee Kwon, Minsoo Jung.

**Data curation:** Minsoo Jung.

**Formal analysis:** Rang Hee Kwon, Minsoo Jung.

**Investigation:** Rang Hee Kwon, Minsoo Jung.

**Software:** Minsoo Jung.

**Writing – original draft:** Rang Hee Kwon, So-Hyun Kim, Minsoo Jung.

**Writing – review & editing:** Rang Hee Kwon, So-Hyun Kim, Minsoo Jung.

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
