## [Decision Letter · Decision Letter 0]

9 May 2023

PONE-D-22-34866Behavioral Predictors Associated with HIV Screening Needs in Gay Korean Men during the COVID-19 PandemicPLOS ONE

Dear Dr. Jung,

Thank you for submitting your manuscript to PLOS ONE. After careful consideration, we feel that it has merit but does not fully meet PLOS ONE’s publication criteria as it currently stands. Therefore, we invite you to submit a revised version of the manuscript that addresses the points raised during the review process.

We look forward to receiving your revised manuscript.

Kind regards,

Omid Dadras, MD, PhD

Academic Editor

PLOS ONE

Journal Requirements:

This work was supported by the National Research Foundation of Korea Grant funded by the Korean Government (NRF-2022R1F1A1062998; PI: Prof. Dr. Minsoo Jung).

Reviewers' comments:

Reviewer's Responses to Questions

**Comments to the Author**

1. Is the manuscript technically sound, and do the data support the conclusions?

Reviewer #1: Partly

Reviewer #2: Yes

Reviewer #3: Yes

2. Has the statistical analysis been performed appropriately and rigorously? 

Reviewer #1: I Don't Know

Reviewer #2: I Don't Know

Reviewer #3: Yes

3. Have the authors made all data underlying the findings in their manuscript fully available?

Reviewer #1: Yes

Reviewer #2: Yes

Reviewer #3: Yes

4. Is the manuscript presented in an intelligible fashion and written in standard English?

Reviewer #1: Yes

Reviewer #2: Yes

Reviewer #3: Yes

5. Review Comments to the Author

Reviewer #1: Good work done. However, I wish you could cite more to support your findings in the discussion sections. Also, structure the strengths and limitations and conclusion sections with headings to make the work standout.

Reviewer #2: The authors are to be congratulated for considering this important subject..The concept of the paper is good, starting from the introduction, methodology, discussion to a conclusion. The result of the paper supports the conclusion.

Reviewer #3: This is a good study reporting behavioral charateristics among MSM during pandemic. This is very importance baseline data to update policy on accessibility or availability of appropriate HIV screening services to MSM during pandemic. Applause to all authors.

6. PLOS authors have the option to publish the peer review history of their article (what does this mean?). If published, this will include your full peer review and any attached files.

Reviewer #1: **Yes: **Stanley Kofi Alor

Reviewer #2: **Yes: **Agmas Wassie Abate

Reviewer #3: **Yes: **Rimah Melati AB Ghani

---

## [Author Response · Author response to Decision Letter 0]

17 May 2023

Reviewers’ Comments

1. This is a good study reporting behavioral characteristics among MSM during pandemic. This is very importance baseline data to update policy on accessibility or availability of appropriate HIV screening services to MSM during pandemic.

>>> Authors’ Response: Thank you for recognizing the needs and strengths of our research. In this revision, the authors supplemented the manuscript to reveal the novelty of the study more clearly.

2. Page 8 line: [The time taken to complete the self-administered questionnaire was approximately 20 minutes.] I would suggest to authors to explain details on self-administered questionnaire in the Methods. Is it validated self-administered questionnaire? Which previous study using this validated self-administered questionnaire? If not using validated self-administered questionnaire, any validation done prior to current study? Example of validated self-administered questionnaire need to upload in the full paper as appendix or supporting document.

>>> Authors’ Response: The survey instruments and self-administered questionnaire of this study were verified for validity and reliability in "Developing the behavioral monitoring survey system for the high HIV risk group", a project supported by the Korea Federation for HIV/Prevention in 2011. At that time, the project aimed to develop behavioral monitoring indicators suitable for LGBTI people in Korea based on UNAIDS' "Core Indicators for National AIDS Programs" in 2007. The author of this paper participated in the project at the time and has been updating the survey tools and questionnaire while carrying out the project on homosexuals in Korea. However, in the past 10 years, project-related data were classified as confidential and could not be published as a academic thesis. Since the relevant information has now been declassified, at the request of the reviewer, we have uploaded the entire questionnaire as a supporting document.

3. Reviewer #1: Good work done. However, I wish you could cite more to support your findings in the discussion sections. Also, structure the strengths and limitations and conclusion sections with headings to make the work standout.

>>> Authors’ Response: Following the reviewer's recommendations, we additionally cited recent literature to support our article in the discussion section. Also, we divided the paragraphs to clearly reveal the strengths and limitations of the thesis.

4. Reviewer #2: The authors are to be congratulated for considering this important subject. The concept of the paper is good, starting from the introduction, methodology, discussion to a conclusion. The result of the paper supports the conclusion.

>>> Authors’ Response: Thank you for praising the value and originality of this study. In this revised manuscript, we highlighted the strengths and supplemented the weaknesses of this study. As a result, we believe that our manuscript is much stronger and clearer.

5. Reviewer #3: This is a good study reporting behavioral characteristics among MSM during pandemic. This is very importance baseline data to update policy on accessibility or availability of appropriate HIV screening services to MSM during pandemic. Applause to all authors.

>>> Authors’ Response: Thank you for praising the value and originality of this study. In this revised manuscript, we highlighted the strengths and supplemented the weaknesses of this study. As a result, we believe that our manuscript is much stronger and clearer.

---

## [Editor Report · Decision Letter 1]

29 May 2023

Behavioral Predictors Associated with HIV Screening Needs in Gay Korean Men during the COVID-19 Pandemic

PONE-D-22-34866R1

Dear Dr. Jung,

We’re pleased to inform you that your manuscript has been judged scientifically suitable for publication and will be formally accepted for publication once it meets all outstanding technical requirements.

Kind regards,

Omid Dadras, MD, PhD

Academic Editor

PLOS ONE

---

## [Editor Report · Acceptance letter]

2 Jun 2023

PONE-D-22-34866R1 

Behavioral Predictors Associated with HIV Screening Needs in Gay Korean Men during the COVID-19 Pandemic 

Dear Dr. Jung:

I'm pleased to inform you that your manuscript has been deemed suitable for publication in PLOS ONE. Congratulations! Your manuscript is now with our production department. 

Kind regards, 

on behalf of

Dr Omid Dadras 

Academic Editor

PLOS ONE